# The Impact of Various Superalloys on the Oxidation Performance of Nanocrystalline Coatings at High Temperatures

Bo Meng [1], Shasha Yang [1], Jing Zhao [2], Jinlong Wang [1,*], Minghui Chen [1] and Fuhui Wang [1]

1 Shenyang National Laboratory for Materials Science, Northeastern University, Shenyang 110819, China; fhwang@mail.neu.edu.cn (F.W.)
2 Nuclear Power Institute of China, Chengdu 528208, China
* Correspondence: wangjinlong@mail.neu.edu.cn; Tel.: +86-24-23904856; Fax: +86-24-23893624

**Abstract:** Nanocrystalline coatings with the same chemical composition as an N5 superalloy were prepared on K38 and N5 superalloys by magnetron sputtering. The effect of different superalloys on the high temperature oxidation behavior of nanocrystalline coatings was investigated through oxidation kinetics, X-ray diffraction (XRD), scanning electron microscopy (SEM), energy dispersive spectrometry (EDS), and wavelength dispersive spectrometry (WDS). The results indicated that K38-N5 had better oxidation resistance than N5-N5 due to the diffusion of Zr from the K38 superalloy into the oxide scale. In addition, no interdiffusion occurred in the K38 superalloy. The formation of Ta-rich phases in the $Al_2O_3$ scale leads to decrease the oxidation resistance of nanocrystalline coatings. However, the presence of Zr inhibits the formation of Ta-rich phases.

**Keywords:** high temperature oxidation; nanocrystalline coatings; interdiffusion; oxidation behavior; reactive element

## 1. Introduction

The constant demand for improving engine efficiency has led to high temperature protective coatings becoming a hot research topic [1–6]. As the most widely used traditional high-temperature protective coatings, aluminide and MCrAlY coatings exhibit excellent oxidation resistance. Nevertheless, they face significant challenges due to interdiffusion caused by the different chemical compositions with superalloys [7–11]. The occurrence of interdiffusion will not only reduce the oxidation resistance of the coating, but also reduce the mechanical properties of the superalloy. As interdiffusion occurs, the concentrations of Ni and Al in the coating increase and decrease, respectively. The increase in Ni concentration and the reduction in Al concentration will transform the β-NiAl into the γ-Ni and the oxidation resistance of γ-Ni is lower than that of β-NiAl [12,13]. Sato et al. [14] investigated the influence of interdiffusion on the high temperature creep performance of superalloys, which were coated with aluminum diffusion coatings. The results indicated that the creep-rupture lives of coated superalloys were shorter than those of the bare superalloys. Therefore, finding a solution to suppress interdiffusion has become a top priority.

Nanocrystalline coatings, a novel type of high-temperature protective coatings, exhibit excellent oxidation resistance due to their unique design in composition and structure [15]. The nanocrystalline coatings have the same chemical composition as the superalloy, inhibiting the occurrence of interdiffusion between them. More importantly, nanocrystalline coatings, prepared by magnetron sputtering, are a columnar nanostructure in which the high density of grain boundaries reduces the critical aluminum concentration for the coating to form a protective oxide scale. The latest development of nanocrystalline coating is applied to the second generation single-crystal nickel-based N5 superalloy. Wang et al. [16] studied the high temperature oxidation behavior of the nanocrystalline coating at 1000 and 1100 °C and compared it with NiCrAlY coatings deposited on the N5 superalloy. The

results showed that the oxidation rate of the nanocrystalline coating was less than half that of the NiCrAlY coating.

However, nanocrystalline coating, which has the same chemical composition as the N5 superalloy, is imperfect. After oxidation, the $\alpha$-$Al_2O_3$ scale on the nanocrystalline coating will be mixed with Ta-rich phases. The coefficient of thermal expansion mismatch between Ta-rich phases and $\alpha$-$Al_2O_3$ will promote the formation of cracks in the oxide scale, accelerating the growth of the oxide scale [16]. In order to solve the harmful effects of the Ta-rich phase, researchers have proposed various solutions. Wang et al. [17] found that the doping of Y could decelerate the formation of Ta-rich phases in the oxide scale at 1050 °C. In addition, the increase in Al content in the coating could also enhance the effect of Y on inhibiting the formation of the Ta-rich phase [18]. Meng et al. [19] also found that oxygen doping could inhibit the formation of this Ta-rich phase at 1100 °C.

In our previous work, we deposited a nanocrystalline coating with the same chemical composition as the N5 superalloy on the K38 superalloy. The results showed no Ta-rich phase in the oxide scale after oxidation at 1050 °C [20]. Compared with Ref. [16], the oxidation mass gain was lower than that of the coating deposited on the N5 superalloy. This indicated that the K38 superalloy influenced the oxidation behavior of the nanocrystalline coating. Therefore, it is of great significance to study the influence of the substrate alloy on the oxidation behavior of the nanocrystalline coating, especially the suppression of Ta-rich phases.

This study deposited nanocrystalline coatings with the same chemical composition as the N5 superalloy on the N5 and K38 superalloys, respectively. The isothermal oxidation of two groups of samples at 1100 °C was investigated in order to study the effect of different superalloys on its oxidation behavior, especially in the Ta-rich phases.

## 2. Materials and Methods

The cast K38 superalloy and single-crystal N5 superalloy were chosen as the substrate alloy, and their chemical compositions are shown in Table 1. Cylindrical specimens of the N5 superalloy were $\Phi$15 mm $\times$ 2 mm, and the size of the K38 samples was $15 \times 10 \times 2$ mm$^3$. Two groups of samples were ground down to 400 grit using SiC paper and sand-blasted with $SiO_2$ under a pressure of 0.4 MPa. Subsequently, samples were degreased using an ultrasonic cleaner in ethanol and acetone. The coatings were deposited by magnetron sputtering deposition. The target used for sputtering was a $382 \times 128 \times 8$ mm$^3$ sheet with the same chemical composition as the N5 superalloy, as shown in Table 1. The sputtering parameters were set as follows: substrate, temperature of 200 °C, sputtering current of 3.5 A, argon pressure of 0.2 Pa, and sputtering duration of 11 h. For convenience, the samples with the K38 and N5 superalloys were denoted as K38(superalloy)-N5(coating) and N5(superalloy)-N5(coating), respectively.

**Table 1.** Chemical composition of the K38,N5 superalloys, and nanocrystalline coating (wt.%).

| | Ni | C | Cr | Co | W | Mo | Al | Ti | Fe | Nb | Ta | Zr | Re |
|---|---|---|---|---|---|---|---|---|---|---|---|---|---|
| Superalloy K38 | Bal | 0.1–0.2 | 15.7–16.3 | 8–9 | 2.4–2.8 | 1.5–2 | 3.2–3.7 | 3.0–3.5 | ≤0.5 | 0.6–1.1 | 1.5–2.0 | 0.05–0.15 | |
| Superalloy N5 | Bal | | 7.0 | 7.5 | 5.0 | 1.5 | 6.2 | | | | 6.5 | | 3.0 |
| Nanocrystalline coating | Bal | | 7.0 | 7.5 | 5.0 | 1.5 | 6.2 | | | | 6.5 | | 3.0 |

Each group specimen with five parallel specimens was placed in alumina crucibles for an isothermal oxidation test. In order to ensure accurate weight measurements of specimens, it was necessary to maintain a constant weight for the crucibles. Therefore, they were heat treated in the furnace at 1250 °C until the quality remained unchanged. Subsequently, specimens were kept at 1100 °C for different times with weight measurements performed at appointed intervals. The weight of the specimens was recorded using an electronic balance with a precision of 0.01 mg (Sartorius BP211D, Göttingen, Germany).

The oxidation test was carried out in a muffle furnace (Luoyang Kere Furnaces Co., LTD., Luoyang, China) in the air. The phase compositions were analyzed using grazing incidence X-ray diffraction (GI-XRD, X' Pert PRO PANalytical Co., Almelo, Holland, CuK$\alpha$

radiation at 40 Kv) with a grazing angle of 0.6°. Surface and cross-sectional morphologies of coatings were observed using a filed-emission scanning electron microscope (FE-SEM, Inspect F50, FEI Co., Ltd, Hillsboro, OR, USA) equipped with an energy-dispersive spectrometer (EDS, X-Max, Oxford Instruments Co., Oxford, UK) and transmission electron microscopy (TEM, JEM-2100F, JEOL, Tokyo, Japan). Elemental mapping of oxidized samples was characterized using an electron probe microanalyzer (EPMA, JEOL 8530F, Japan) equipped with a wavelength dispersive spectrometer (WDS).

## 3. Results

### 3.1. Microstructure of the as-Deposited Nanocrystalline Coatings

Figure 1 shows the morphologies of the as-deposited K38-N5. As shown in Figure 1a, it can be observed that the nanocrystalline coatings exhibit a uniform thickness of approximately 30 µm; Figure 1b is the microstructure of the nanocrystalline coating via TEM. The nanocrystalline coating consists of a columnar nanostructure with a grain width of about 100 nm. Since the coating deposited on the N5 superalloy is the same as the coating deposited on the K38 superalloy, the description of N5-N5 will not be repeated.

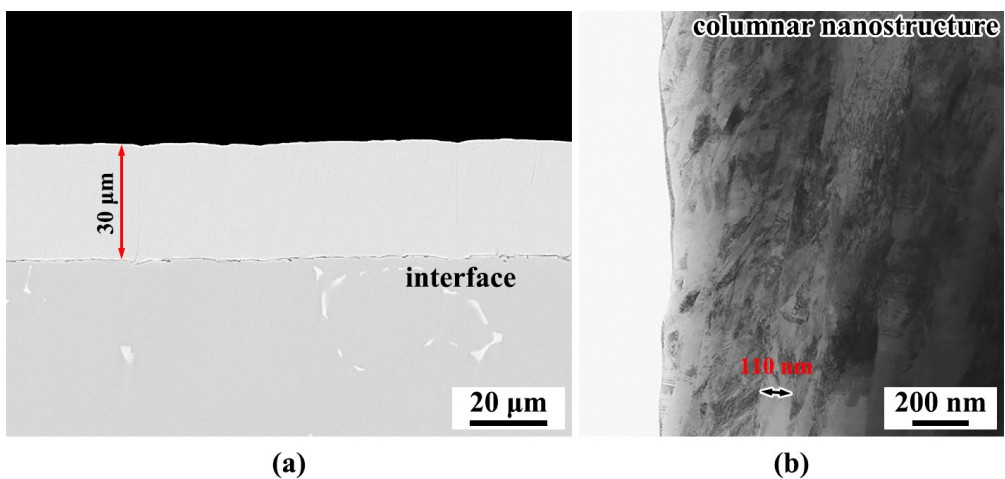

**(a)** **(b)**

**Figure 1.** The morphologies of the as-deposited nanocrystalline coating: (**a**) cross-sectional morphologies; (**b**) columnar nanostructure of nanocrystalline coating.

### 3.2. Oxidation Kinetics

Figure 2 illustrates the kinetic curves of K38-N5 and N5-N5 at 1100 °C for 100 h. The linear interconnection between points is expressed as the changing trend of oxidative weight gain. In Figure 2a, it is evident that the mass gains of K38-N5 are lower than that of N5-N5 throughout the entire oxidation process. During the initial 5 h of oxidation, both K38-N5 and N5-N5 exhibit a relatively large mass gain, and their mass gain is 0.24103 mg·cm$^{-2}$ and 0.31969 mg·cm$^{-2}$, respectively. During the initial 5h of oxidation, the mass gain of K38-N5 is greater than the mass gain from 5 to 100 h. However, the mass gain of N5-N5 is smaller than the mass gain from 5 to 100 h., which shows that the oxidation rate of K38-N5 is lower than that of N5-N5 for 5–100 h. During the oxidation for 5–100 h, the oxidation rate of the two coatings decreased compared with the oxidation rate of the initial 5 h. When oxidized to 100 h, the mass gains of K38-N5 and N5-N5 are 0.44273 mg·cm$^{-2}$ and 0.75274 mg·cm$^{-2}$, respectively. The mass gain of N5-N5 after oxidation for 100 h is relatively close to that reported in Ref. [16]. This shows that the K38 superalloy can indeed improve the oxidation resistance of the nanocrystalline coating.

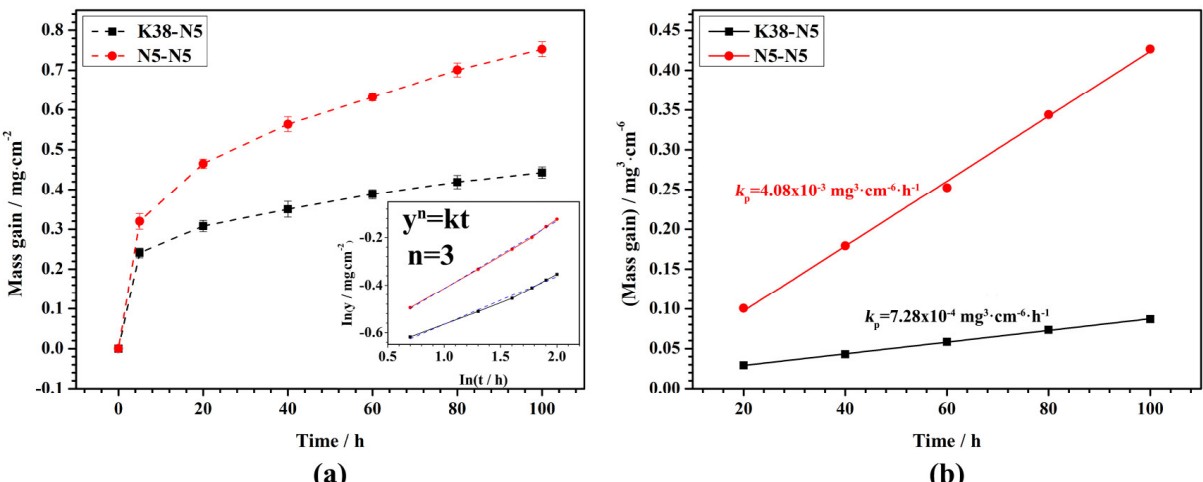

**Figure 2.** The kinetic curves of the nanocrystalline coatings at 1100 °C for 100 h: (**a**) mass gain curve; (**b**) cube of gain curve.

Figure 2a shows a function relationship between log(y) and log(t), where y denotes weight gain and t is oxidation time, which helps to speculate the oxidation regulation of the nanocrystalline coatings. As shown in the figure, the slopes of the two curves, log(y) vs. log(t), were of almost the same value and empirical exponent n = 3. The oxidation rate constants of the two group samples are calculated as follows [17]:

$$y^n = k_p t \tag{1}$$

where $y$ and $t$ denote mass gain and oxidation time, respectively. Hence, the cubic function representing the mass gains over oxidation time demonstrates an approximately linear relationship, as illustrated in Figure 2b. According to Figure 2b, the oxidation rate constants ($k_p$) of K38-N5 and N5-N5 are $7.28 \times 10^{-4}$ $mg^3 \cdot cm^{-6} \cdot h^{-1}$ and $4.08 \times 10^{-3}$ $mg^3 \cdot cm^{-6} \cdot h^{-1}$, respectively. It proves that the oxidation rate of K38-N5 is lower than that of N5-N5.

*3.3. Oxidation Behavior*

The GI-XRD results of the oxide scales formed on K38-N5 and N5-N5 after oxidation for 100 h at 1100 °C are presented in Figure 3. As the oxide scale is relatively thin, the strongest peak observed corresponds to the γ and γ′ phases. After oxidation for 5 h, the oxide scales formed on the two groups of samples are mainly composed of α-Al$_2$O$_3$. During oxidation for 5–100 h, the oxide scales formed on K38-N5 are still composed of α-Al$_2$O$_3$, as shown in Figure 3a. However, Ta-rich oxide appears in the oxide scale formed on N5-N5 after oxidation for 40 h. According to the XRD pattern in Figure 3b, the Ta-rich oxide is Ta$_2$O$_5$. After oxidation for 100 h, the peak of the Ta$_2$O$_5$ becomes stronger than that at 40 h, which proves that the content of the Ta-rich phase in the oxide scale increases continuously with the extension of oxidation time.

Figure 4 displays the morphologies of the oxide scale formed on K38-N5 and N5-N5 at 1100 °C for 100 h. The surface morphologies show that the oxide scale formed on K38-N5 and N5-N5 is complete and without spallation, as shown in Figure 4a,b. The cross-sectional morphologies also confirm that the oxide scales are complete. In Figure 4b, it is clear that the thickness of the oxide scale formed on K38-N5 is 2.3 μm, which is approximately three-fifths of the thickness of the oxide scale formed on N5-N5 (3.5 μm). It is consistent with the oxidation kinetics. Different from the oxide scale formed on N5-N5, oxide pegging can be observed in the oxide scale formed on K38-N5. More importantly, no interdiffusion zone (IDZ) and topologically closed-packed (TCP) phases were formed in the K38-N5 and N5-N5 superalloys.

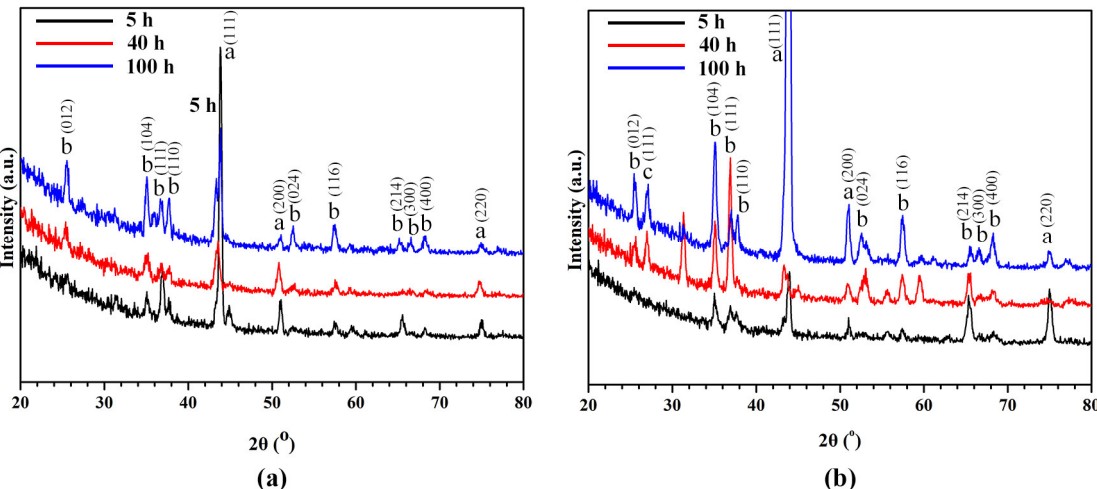

**Figure 3.** GI-XRD patterns after oxidation for 100 h at 1100 °C: (**a**) K38-N5; (**b**) N5-N5 (a. $\gamma/\gamma'$, b. $\alpha$-Al$_2$O$_3$, c. Ta$_2$O$_5$).

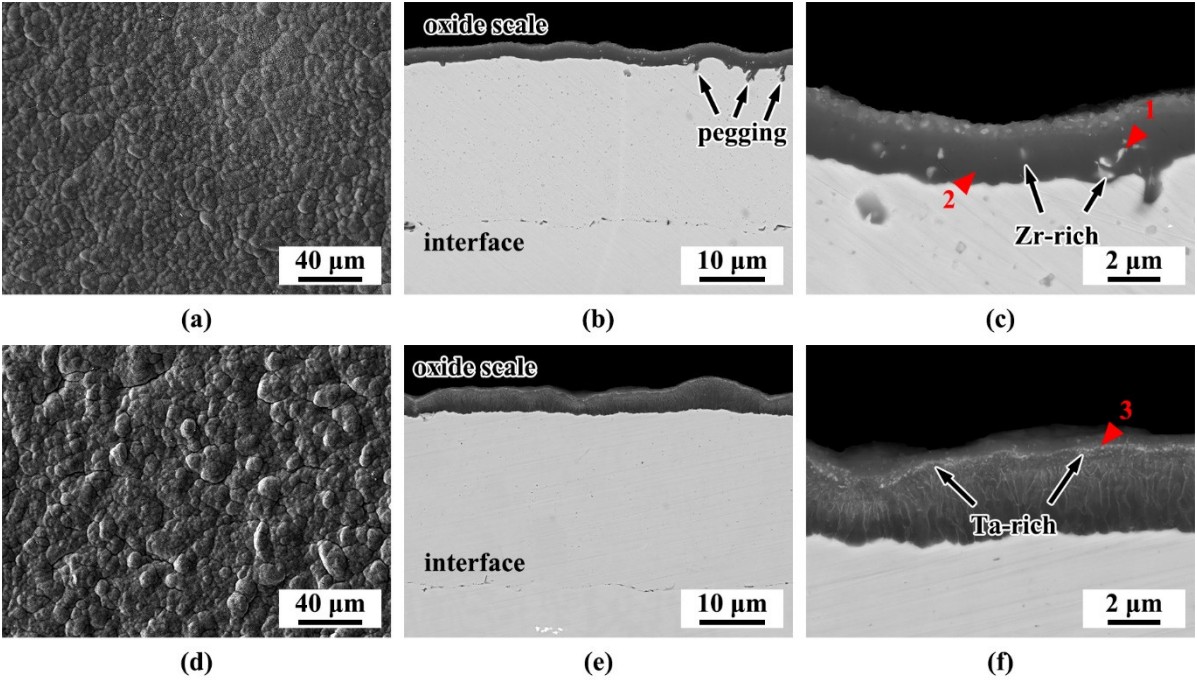

**Figure 4.** Surface and cross-sectional morphologies of K38-N5 and N5-N5 after oxidation for 100 h at 1100 °C: (**a**) surface morphology of K38-N5; (**b**) cross-sectional morphology of K38-N5; (**c**) the high magnification morphology of (**b**) and the marking of EDS point 1, 2; (**d**) surface morphology of N5-N5; (**e**) cross-sectional morphology of N5-N5; (**f**) the high magnification morphology of (**e**) is the marking of EDS point 3.

In order to observe more details of the oxide scales, the cross-sectional morphologies of the oxide scales are magnified, as depicted in Figure 4c,f. In Figure 4c, the oxide scale formed on K38-N5 is divided into inner and outer layers, and a small amount of bright white particles are mixed in the internal layer oxide scale. According to EDS in Table 2, the oxide scale is mainly composed of Al$_2$O$_3$, and the white particles are Zr-rich phases. In addition, a large number of white particles exist in the outer oxide scale but not the Zr-rich phase. The oxide scale formed on the surface of N5-N5 is also mixed with white particles, which is a Ta-rich oxide obtained by EDS. Combined with the XRD results, it can

be inferred that the white phase is Ta$_2$O$_5$. Thus, the white phases mixed in the oxide scale formed on K38-N5 and N5-N5 are Zr-rich phases and Ta-rich phases, respectively.

**Table 2.** EDS-detected compositions for K38-N5 and N5-N5 after oxidation in Figure 4 (wt.%).

| Elements | | O | Al | Ta | Zr | Other |
|---|---|---|---|---|---|---|
| K38-N5 | Point 1 | 36.3 | 45.7 | | 17.3 | 0.7 |
| | Point 2 | 54.2 | 44.9 | | | 0.9 |
| N5-N5 | Point 3 | 45.0 | 44.9 | 8.2 | | 1.9 |

Figure 5 illustrates the elemental mapping of the oxide scale formed on K38-N5 after oxidation for 100 h at 1100 °C. According to the results of WDS and XRD patterns, the oxide scale formed on K38-N5 can be determined to be the α-Al$_2$O$_3$ scale. Although the oxide scale is mixed with Zr-rich oxide, it does not destroy the protective properties of the α-Al$_2$O$_3$ scale. Furthermore, the Zr-rich phases do exist in the oxide pegging. It is worth noting that many white phases near the surface in the oxide scale are Ta-rich phases.

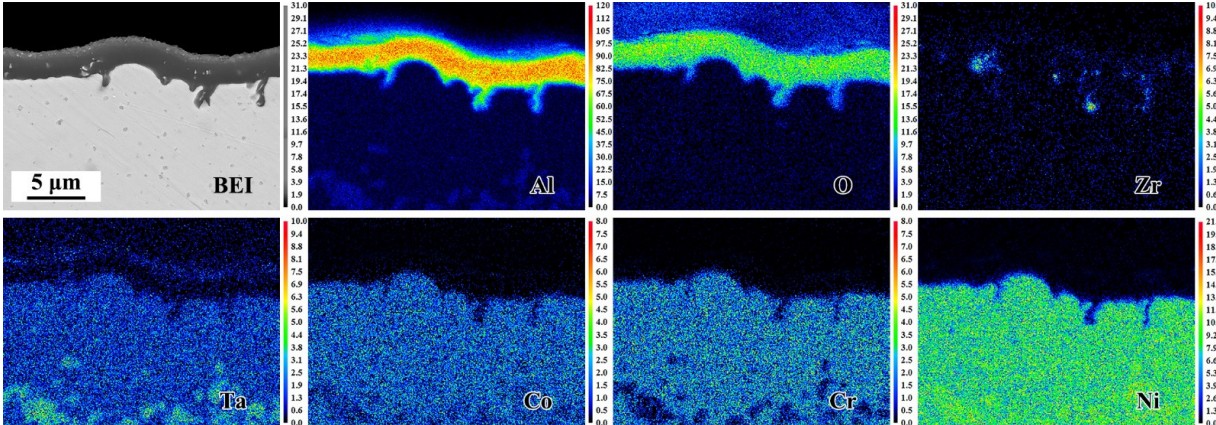

**Figure 5.** Elemental mappings of the oxide scale formed on K38-N5 by WDS.

Figure 6 presents the elemental mapping of the oxide scale formed on N5-N5 after oxidation for 100 h at 1100 °C. Similar to K38-N5, the oxide scale formed on N5-N5 can also be determined to be the α-Al$_2$O$_3$ scale. However, Ta is distributed in the Al$_2$O$_3$ scale homogeneously, which is consistent with our previous reported work [16]. More importantly, the concentration of Ta in the oxide scale near the surface is higher than that in other regions, similar to the outer oxide scale formed on K38-N5.

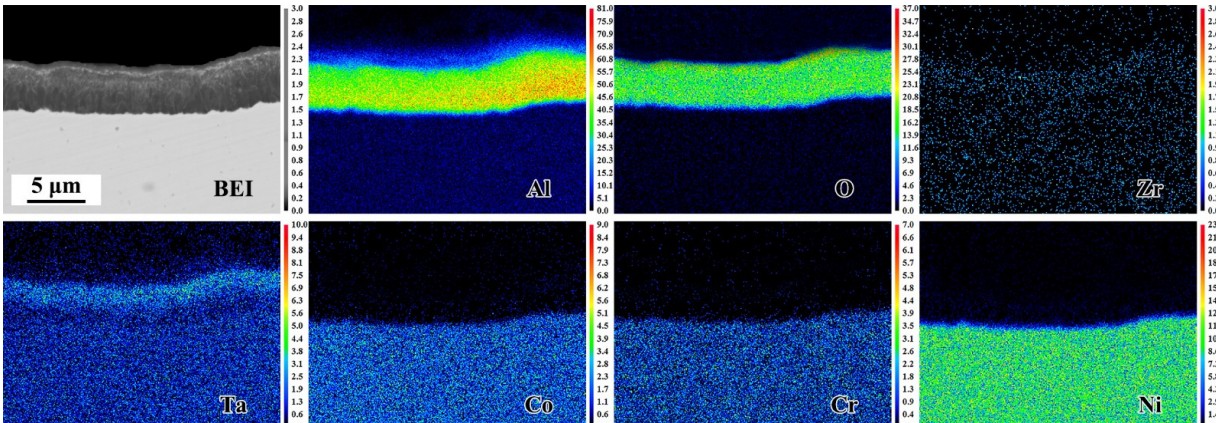

**Figure 6.** Elemental mapping of the oxide scale formed on N5-N5 by WDS.

Interdiffusion between the coating and the superalloy is also an important indicator for evaluating the oxidation resistance of the coating. Figure 7 shows the cross-sectional morphology of the interface between the K38 superalloy and the nanocrystalline coating, and its corresponding elemental mapping by WDS after oxidation for 100 h at 1100 °C. According to the elemental mapping, it can be found that the elements are evenly distributed in the superalloy, except for Al, Cr, and Ta. There is an Al-poor zone about 5 μm in the superalloy, but this zone is rich in Cr.

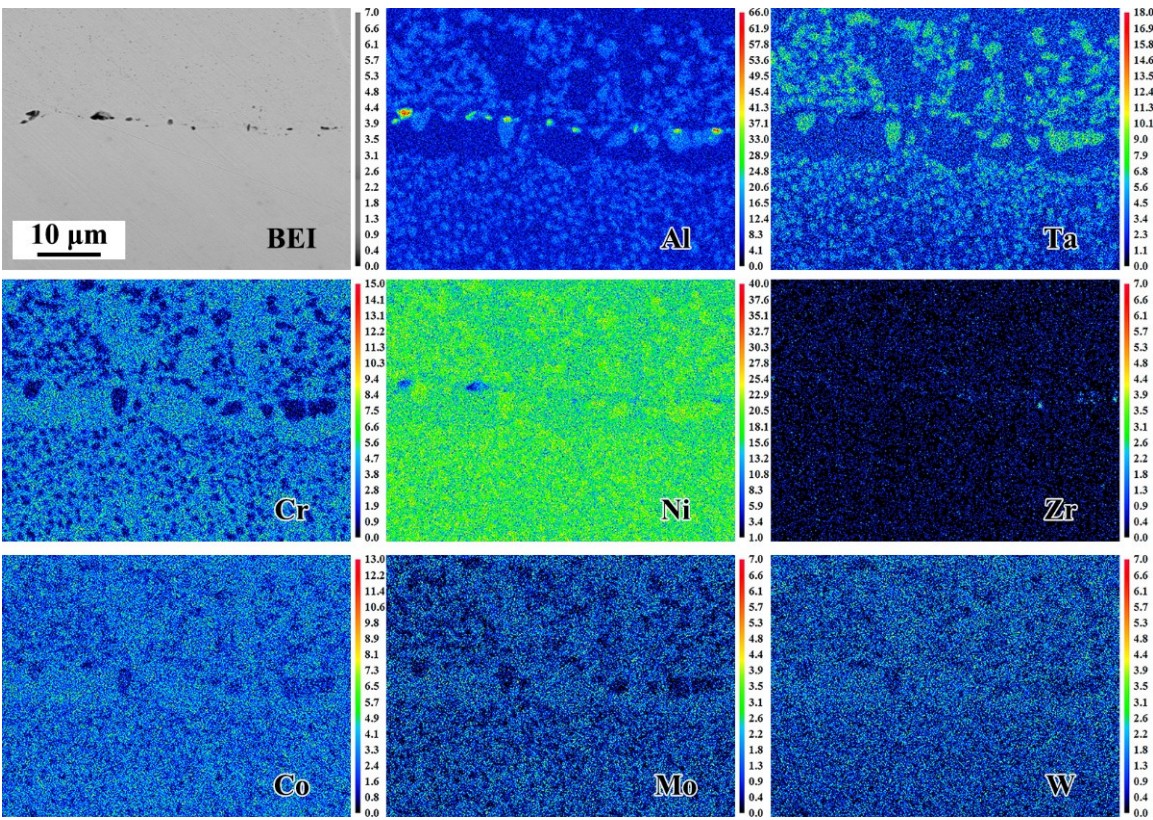

**Figure 7.** Elemental mapping of the interface between K38 superalloy and nanocrystalline coating by WDS.

To understand the formation process of oxide scale, oxide scales formed after oxidation for 5 h and 40 h have been observed.

Figure 8 shows the cross-sectional morphologies of K38-N5 at 1100 °C after a short time. In Figure 8a,c, it can be observed that no IDZ and TCP phases exist throughout the entire oxidation process. Similarly, the oxide scales are divided into inner and outer layers. After oxidation for 5 h, the thickness of the oxide scale formed on K38-N5 is approximately 1.1 μm, and the outer layer oxide scale with Ta-rich is about 0.6 μm. Interestingly, the thickness of the external oxide scale is also about 0.6 μm after oxidation for 40 h, even though the oxide scale thickens to 1.7 μm. However, the inner oxide scale is pure $Al_2O_3$ after oxidation for 5 h, which is different from the oxide scale formed after oxidation for 40 h and 100 h. In addition, when the oxidation reaches 40 h, a Zr-rich phase appears in the inner oxide scale, which is the same as the oxide scale after oxidation for 100 h.

Figure 9 shows the cross-sectional morphologies of N5-N5 at 1100 °C after a short time. After oxidation for 5 h, there are a large number of Ta-rich phases in the oxide scale, as shown in Figure 9b. With the oxidation for 40 h, the Ta-rich phase in the oxide scale becomes more obvious. Compared with the oxide scales formed on K38-N5, the oxide scales formed on N5-N5 are always thicker after oxidation for the same time.

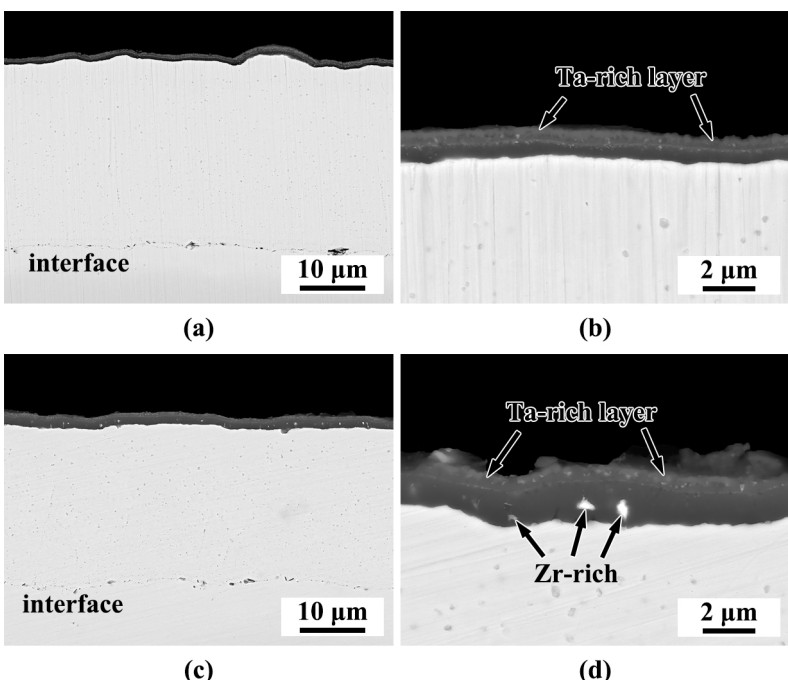

**Figure 8.** Cross-sectional morphologies of K38-N5 after oxidation at 1100 °C for different times: (**a**) 5 h; (**b**) the high magnification morphology of (**a**); (**c**) 40 h; (**d**) the high magnification morphology of (**c**).

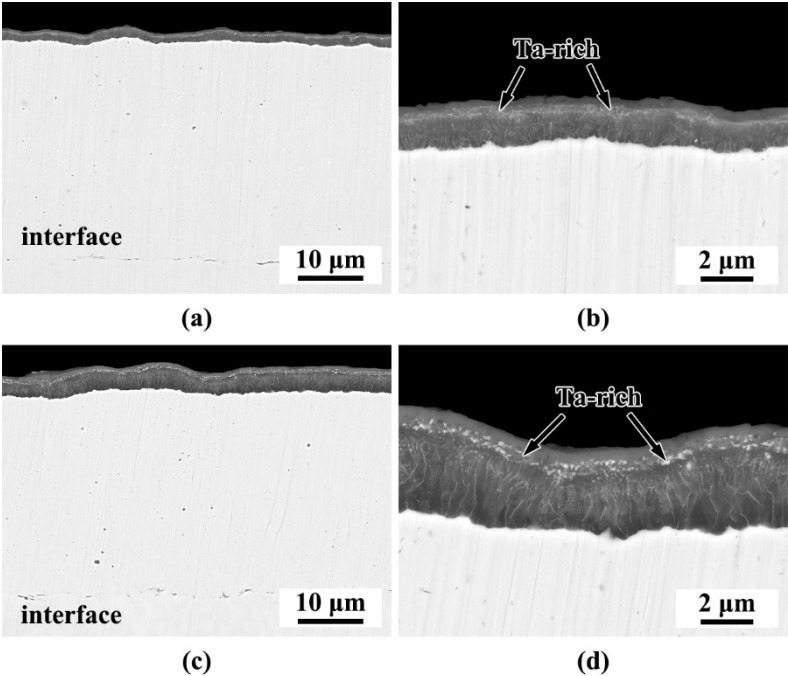

**Figure 9.** Cross-sectional morphologies of N5-N5 after oxidation at 1100 °C for different times: (**a**) 5 h; (**b**) the high magnification morphology of (**a**); (**c**) 40 h; (**d**) the high magnification morphology of (**c**).

## 4. Discussion

Nanocrystalline coatings have excellent oxidation resistance and are attractive worldwide due to their columnar nanostructure [15–19]. According to Wagner's theory [21,22], grain refinement can reduce the critical concentration of Al required to form an $Al_2O_3$ scale on the high-protective coating. However, there are a large number of Ta-rich phases in the oxide scale formed on nanocrystalline coatings with the same chemical composition as the

N5 superalloy. Notably, the formation of Ta-rich phases is suppressed in the oxide scale formed on K38-N5. Details on the mechanism of oxidation behavior and the suppression of Ta-rich phases are discussed as follows:

### 4.1. The Oxidation Behavior of Nanocrystalline Coatings

Wang et al. [16] have reported the oxidation of nanocrystalline coatings at 1100 °C. After oxidation for 100 h, the mass gain reached about 0.8 mg·cm$^{-2}$. This is relatively close to our results from N5-N5. After oxidation, an α-Al$_2$O$_3$ scale would be formed on the nanocrystalline coating for K38-N5 and N5-N5. However, the oxide scale formed on N5-N5 was mixed with Ta-rich phases, which would reduce the oxidation resistance of the nanocrystalline coating [20]. The formation of Ta-rich phases is mainly due to the strong affinity between Ta and O. In Figure 10, the Ellingham diagram provides insight into the thermodynamic behavior of Al, Ta, Cr, Co, and Ni. It could be found that the affinity of Ta to O is only weaker than that of Al. Nevertheless, Ta-rich oxides were also formed because the concentration of Ta (6.5 wt.%) was higher than that of Al (6.2 wt.%) in the coating. Furthermore, as the oxidation time prolonged, the concentration of Al in the coating decreased and Ta was more easily oxidized. Thus, a large number of Ta-rich phases were formed during the growth of the Al$_2$O$_3$ scale.

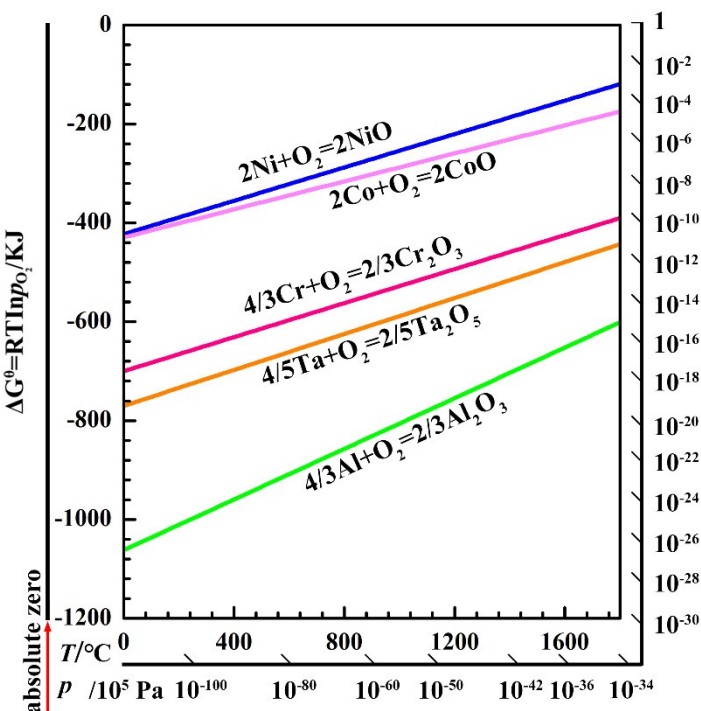

**Figure 10.** Ellingham diagram of Al, Ta, Cr, Co, and Ni (per mole O$_2$).

Different from N5-N5, there were very few Ta-rich phases in the α-Al$_2$O$_3$ scale formed on K38-N5, especially since there were no Ta-rich phases in the inner oxide scale. Therefore, K38-N5 has a lower oxidation rate compared to N5-N5. However, there were Ta-rich phases in the outer oxide scale, even after oxidation for only 5 h. It is worth noting that as the oxidation time increased, the thickness of the outer oxide scale remained unchanged. However, Zr-rich particles appeared in the inner oxide scale after oxidation for 40 h. Since there was no Zr in the nanocrystalline coating, the Zr in the oxide scale was diffused from the K38 superalloy.

More importantly, no interdiffusion occurred between the nanocrystalline coating and the K38 superalloy, even though the chemical compositions of the two were different. The nanocrystalline coating was mainly composed of the supersaturated γ phase, which will maintain phase equilibrium with the K38 superalloy. Sato et al. [23] reported that coatings

formed of $\gamma$ and $\gamma'$ phases could achieve phase equilibrium with the superalloy, which was composed of $\gamma/\gamma'$. However, an Al-poor zone of about 5 μm appeared in the K38 superalloy. The main reason is that Al diffuses into the coating because a large amount of Al is consumed to form an $Al_2O_3$ scale. In addition, the formation of the Al-poor zone also led to a higher concentration of Cr in this zone.

Therefore, the oxidation resistance of K38-N5 was significantly more substantial than that of N5-N5 due to the absence of Ta-rich oxide in the outlay oxide scale. No interdiffusion of K38-N5 resulted from phase equilibrium between the nanocrystalline coating and the K38 superalloy.

### 4.2. The Influence of Reactive Element Zr in the Superalloy K38

The doping of reactive elements could improve the high-temperature oxidation resistance of high-temperature protective coatings or alloys [17,24–33]. Wang et al. [17] reported that doping the reactive element Y could improve the oxidation resistance of nanocrystalline coatings and inhibit the formation of Ta-rich oxides in the oxide scale at 1050 °C. Duan et al. [24] doped Hf into a NiCoCrAlY coating, which increased its lifetime from 240 h to 2664 h. Similarly, the presence of Zr played a crucial role in enhancing the oxidation resistance of nanocrystalline coatings.

On the one hand, Zr suppresses the formation of Ta-rich phases in the inner oxide scale. Yang et al. [18] found that the reactive element Y could inhibit the formation of Ta-rich phases in the oxide scale. As a reactive element, Zr also had a similar influence. In the initial 5 h of oxidation, Ta diffused into the oxide scale and was oxidized because there was no Zr in the nanocrystalline coating. Thus, Ta-rich oxide existed in the outer oxide scale. Subsequently, Zr in the K38 superalloy diffused into the nanocrystalline coating, inhibiting the continued formation of Ta-rich oxides. As a result, the Ta-rich phases only existed in the outer scale, and the thickness of the external oxide scale remained unchanged regardless of the oxidation time.

On the other hand, the reactive element could improve the adhesion of the oxide scale due to the formation of oxide pegging [34–40]. Figure 4b found that oxide pegging nailed into the nanocrystalline coating after oxidation for 100 h. More importantly, the presence of Zr-rich oxides could be observed in oxide pegging, as shown in Figure 5. The formation of oxide pegging was mainly due to the diffusion of Zr to the grain boundary of the $Al_2O_3$ scale, which accelerated the inward diffusion of O, resulting in protruding oxides nailed into the nanocrystalline coating. Mennicke et al. [41] proved that the formation of oxide pegging could prevent the propagation of cracks at the interface between the oxide scale and coating. Thus, Zr in the K38 superalloy would diffuse into the oxide scale, which could improve its adhesion.

To conclude, Zr not only improves the oxide scale adhesion but also suppresses the formation of the Ta-rich phase.

### 5. Conclusions

From the above study, the following conclusions can be drawn:

(1) The K38-N5 had better high-temperature oxidation resistance than that of N5-N5. The presence of the Ta-rich phase accelerates the oxidation rate of N5-N5;
(2) No interdiffusion occurred in K38-N5. The phase equilibrium between the nanocrystalline coating and the K38 superalloy suppresses interdiffusion.
(3) The presence of Zr enhanced the oxidation resistance of K38-N5 and improved the adhesion of the oxide scale.

**Author Contributions:** Conceptualization, B.M., J.W. and M.C.; Methodology, B.M., J.W., S.Y., J.Z. and M.C.; Investigation, B.M., J.Z. and J.W.; Data curation, B.M., J.Z. and S.Y.; Writing—original draft preparation, B.M. and J.W.; Writing—review and editing, B.M., J.W., S.Y., M.C. and F.W.; Visualization, J.W., S.Y., M.C. and F.W. All authors have read and agreed to the published version of the manuscript.

**Funding:** This project is financially supported by the National Natural Science Foundation of China under Grant (51671053 and 51801021), the Fundamental Research Funds for the Central Universities (No. N2302007), and the Ministry of Industry and Information Technology Project (No. MJ-2017-J-99).

**Institutional Review Board Statement:** Not applicable.

**Informed Consent Statement:** Not applicable.

**Data Availability Statement:** Not applicable.

**Conflicts of Interest:** The authors declare that they have no known competing financial interest or personal relationships that could have appeared to influence the work reported in this paper.

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
