# Peer review of "The Impact of Various Superalloys on the Oxidation Performance of Nanocrystalline Coatings at High Temperatures"

_coatings, doi:10.3390/coatings13101770_

Round 1

Reviewer 1 Report

The manuscript discusses the high-temperature oxidation behavior of K38 and N5 with the nanocrystalline coatings. XRD, EDS, SEM-EDS and WDS were utilized to characterize the samples. The manuscript seems to be interesting and can be accepted after addressing the following comments:

1)  There are many grammatical errors throughout the manuscript. 

2) The abstract must contain important findings to increase the interest of the readers.

3) Insert the reference for the equation. 1.

4) The caption of Fig.3 is wrong.

5) Fig.4 shows the presence of more Al and O content, not Zr content, which is against the author's claim. 

6) How does the Ta affect the overall oxidation behavior of K38 and N5?

7) Can you compare your results with the literature if possible?

English is adequate

Author Response

Thank you for your letter and your useful suggestions. We appreciate the reviewers' constructive feedback on our manuscript titled " The Impact of Various Superalloys on the Oxidation Performance of Nanocrystalline Coatings at High Temperatures", which was revised carefully according to both the reviewers' and the editor's comments and suggestions. The main revisions are highlighted in red color in the revised manuscript and the following is a point-to-point response to your comments. Please refer to the attachment below.

Reviewer 2 Report

The present manuscript deals with the high temperature oxidation behavior of nanocrystalline coatings based on superalloys. It is well written and discussed. However, some major revision is mandatory before it can be considered for publication. The main comments are given below:

1. What is the meaning of the “SN” abbreviation? It should be either explained or removed from the denotation as it seems not to bring any information and is rather confusing.

2. The sentence “Because the phase equilibrium between the nanocrystalline coating and superalloy K38 could inhibit the occurrence of interdiffusion.” is useless in Abstract.

3. The text between lines 47 and 52 should be cited. It is not clear whether the findings come from authors or from literature.

4. The sentence in lines 78 and 79 “After coating deposition, the thickness of the coating was approximately 30 μm.” should be moved to the Results section. Possibly, the coating thickness could be marked in the SEM images.

5. The equipment used for the isothermal exposure should be specified (furnace type, technical air flow rate vs “natural” atmosphere, etc.).

6. Fig. 1a – scientifically, the linear interconnection of measured points is not correct. The authors cannot be sure that for example after 10 hours the measured value lies in the linear line. The approximation should be used instead.

7. What grazing angle was used in GI XRD? It should be specified in Materials and Methods.

8. The interpretation of Fig. 2 is confusing. The legend should be included to distinguish the black, red and blue pattern. Moreover, the Miller indices of phases are missing and should be added.

9. It would be appropriate to mention EDS points also in Fig. 3 caption. Besides that, K38 and N5 are not distinguished in the figure caption probably by mistake.

10. Why did not the authors use more precise WDS instead of EDS to measure the point chemical composition given in Table 2?

11. The abbreviations IDZ and TCP should be explained in the text.

12. The obtained oxidation rates should be compared to literature in the discussion.

13. The manuscript also needs some minor formal editing:

- line 13 – diffractometer ---> diffraction

- line 14 – spectrometer ---> spectrometry

- line 92 – file emission ---> field emission

- line 128 – oxides ---> oxide

English language is fine, minor editing is required.

Author Response

(The authors gave the same response as above.)

Reviewer 3 Report

The introduction is to be rewritten in a technical manner.

The XRD patterns are fine but the discussion on structure is not up to the mark.

How can you identify the oxide in the microstructure?

Provide a scheme for synthesis.

Explain cast superalloy and single-crystal superalloy.

Provide the novelty and motivation behind the work.

minor

Author Response

(The authors gave the same response as above.)

Reviewer 4 Report

In this paper, the oxidation kinetics of two Ni-based superalloys K38 and N5 has been studied. The alloys were coated with N5 coating. The oxidation kinetics was studied by discontinuous weight gain measurements at 1100 °C. The maximum annealing time was 100 h. The oxide scale was inspected by SEM/EDS and XRD. Although the paper has some merits, the materials characterization and discussion are insufficient to warrant publication. Furthermore, the paper lacks a rationale behind the choice of materials. As such, I cannot recommend the paper for publication in its current form. The following comments should be considered:

1.The paper describes the coatings as “nanocrystalline” (see the title); however, there is no justification for it. You have to show the nanocrystallinity by studying the materials via HR TEM/SAED.

2.The paper is written in poor English. As such, it is difficult to follow. The authors obviously claim that the substrate alloy can have an effect on the oxidation behavior of the coatings (see lines 11-12). However, it is the opposite; i.e., the coating can retard the oxidation of the substrate, not vice versa. The authors have to substantially rewrite their paper and improve their English. It is advised to approach a native speaker familiar with Materials Science to proof-read the paper.

3.The authors coated the N5 alloy with a N5 coating. I don’t see the logic behind it. Is the alloy coated with the same material? Why?

4.There are no weight gain data for the K38 substrate. They must be included. The same applies to the N5 alloy substrate. They must be included for the sake of comparison.

5.The oxidation kinetics is analyzed via cubic law (Fig. 1). However, there is no justification for this. Usually, a parabolic rate law is used. You have to explain why you think the cubic law is valid in this case.

6.The oxidation atmosphere is not specified. Was it air, oxygen, or something else?

7.The paper cites a recent paper by the authors (https://doi.org/10.3390/coatings10121188) in which the same materials were oxidized at 1050 °C. However, the oxidation rate constant for the N5-SN reported at 1100 °C is lower than the rate constant at 1050 °C. It contradicts the Arrhenius law. You have to explain it.

8. The rate constant of K38-SN at 1100 ° C is lower compared to N5-SN; however, the opposite is observed at 1050 °C (https://www.mdpi.com/coatings/coatings-10-01188/article_deploy/html/images/coatings-10-01188-g002.png). It raises several questions regarding the reliability of your results. Why is the behavior at 1100 °C completely different compared to 1050 °C?

9.How thick were the coatings?

10.There is a Ta depleted layer below the oxide scale of K38-SN (see Fig. 4). The oxidation mechanism of Ta needs to be sufficiently explained and discussed.

11.Powder diffraction file numbers of phases identified via XRD should be given in the legend of Fig. 2. Explain the difference between the blue, red, and black diffractograms in the figure. Are they measured at different oxidation times?

The paper is written in poor English. As such, it is difficult to follow. The authors obviously claim that the substrate alloy can have an effect on the oxidation behavior of the coatings (see lines 11-12). However, it is the opposite; i.e., the coating can retard the oxidation of the substrate, not vice versa. The authors have to substantially rewrite their paper and improve their English. It is advised to approach a native speaker familiar with Materials Science to proof-read the paper.

Author Response

(The authors gave the same response as above.)

Round 2

Reviewer 2 Report

The authors reflected most of the comments in the revised manuscript. However, the main issues are still current Therefore, the major revision is mandatory. The comments are following (the numbering follows the authors’ responses):

Point 6: I do not agree with the scientific correctness of the linear interconnection. Unfortunately, it is often used by many research groups in the world. The probability of the possible measured value of mass gain after 10 h of oxidation lying on that linear line between 0 and 20 h is very low. In the case of approximation function instead of the linear interconnection, the probability increases and this approach is more correct. I understand that the authors focus on their measured values only, but it should be looked at from the perspective. Eventually, dashed line, but with the proper explanation, can be used instead of the solid line.

Point 8: The presence of Miller indices in XRD patterns proves the correct phase analysis. Since the authors have assigned phases to individual maxima, it should not be a problem to index them. In a scientific paper, Miller indices should be part of basic phase analysis regardless of the discussion and regardless of the customs in the given research field.

English language is fine, minor editing is required

Author Response

Thank you for your letter and your useful suggestions. We appreciate the reviewers' constructive feedback on our manuscript titled " The Impact of Various Superalloys on the Oxidation Performance of Nanocrystalline Coatings at High Temperatures", which was revised carefully according to both the reviewers' and the editor's comments and suggestions. The main revisions are highlighted in red color in the revised manuscript and the following is a point-to-point response to your comments:

Point 6: I do not agree with the scientific correctness of the linear interconnection. Unfortunately, it is often used by many research groups in the world. The probability of the possible measured value of mass gain after 10 h of oxidation lying on that linear line between 0 and 20 h is very low. In the case of approximation function instead of the linear interconnection, the probability increases and this approach is more correct. I understand that the authors focus on their measured values only, but it should be looked at from the perspective. Eventually, dashed line, but with the proper explanation, can be used instead of the solid line.

Response: Thanks for your kind advice. We have been deeply inspired by your scientific correctness of linear interconnection, especially in our study of kinetic curves. We have modified Figure 2 and added relevant instructions in the manuscript.

Point 8: The presence of Miller indices in XRD patterns proves the correct phase analysis. Since the authors have assigned phases to individual maxima, it should not be a problem to index them. In a scientific paper, Miller indices should be part of basic phase analysis regardless of the discussion and regardless of the customs in the given research field.

Response: Thank you for your important suggestion. We have added the Miller indices in XRD patterns.

Reviewer 4 Report

The authors partially answered my previous comments and improved their paper. It is acceptable subject to revision.

1.The dimension of the arrow in Fig. 1b should be 100 nm instead of 100 μm if the coating was nanocrystalline.

2.The cubic law is not confirmed. You must plot the logarithm of weight gain versus the logarithm of time and show it in the paper. Only from the slope of the plot, you can evaluate the kinetic exponent (1, 2, 3, etc.) and judge whether the linear, parabolic, cubic, or other law is valid.

3.In the discussion, the authors say in lines 242-243: “After oxidation for 100 h, the mass gain reached about 0.8 mg·cm-2 and the Kp is 6.48 x 10-4 mg2·cm-4·h-1. This is relatively close to our results of N5-N5.” However, these data are not comparable. You cannot compare the parabolic rate constant with the cubic rate constant. You have not obtained any parabolic rate constant here. See also the comment above.

4.In Fig. 8d you indicated some Y-rich particles. Where did yttrium come from? There is no yttrium in Table 1 (chemical composition of the materials).

Author Response

Thank you for your letter and your useful suggestions. We appreciate the reviewers' constructive feedback on our manuscript titled " The Impact of Various Superalloys on the Oxidation Performance of Nanocrystalline Coatings at High Temperatures", which was revised carefully according to both the reviewers' and the editor's comments and suggestions. The main revisions are highlighted in red color in the revised manuscript and the following is a point-to-point response to your comments:

Point 1: The dimension of the arrow in Fig. 1b should be 100 nm instead of 100 μm if the coating was nanocrystalline.

Response: Thank you for bringing the mistake to our attention. We have modified Figure 1 and modified the description related to the Figure 1. Nanocrystalline coating consists of a columnar nanostructure with a grain width of about 100 nm.

Point 2: The cubic law is not confirmed. You must plot the logarithm of weight gain versus the logarithm of time and show it in the paper. Only from the slope of the plot, you can evaluate the kinetic exponent (1, 2, 3, etc.) and judge whether the linear, parabolic, cubic, or other law is valid.

Response: Thank you for suggesting ways to improve the accessibility of our manuscript. We have plotted the logarithm of weight gain versus the logarithm of time and shown it in Figure 2a. The cube of mass weight linearly with time can also prove that the oxidation of nanocrystalline coatings in this paper follows cubic law.

Point 3: In the discussion, the authors say in lines 242-243: “After oxidation for 100 h, the mass gain reached about 0.8 mg·cm-2 and the Kp is 6.48 x 10-4 mg2·cm-4·h-1. This is relatively close to our results of N5-N5.” However, these data are not comparable. You cannot compare the parabolic rate constant with the cubic rate constant. You have not obtained any parabolic rate constant here. See also the comment above.

Response: Thanks for your important suggestion. We indeed cannot compare the parabolic rate constant with the cubic rate constant. In fact, weight gain is also the most intuitive data reflecting the oxidation rate. The weight gain reported in the literature is very close to the results in our experiments, both close to 0.8 mg·cm-2 after 100 h of oxidation. Therefore, we have removed the comparison of the rate constant in the revised manuscript.

Point 4: In Fig. 8d you indicated some Y-rich particles. Where did yttrium come from? There is no yttrium in Table 1 (chemical composition of the materials).

Response: Thank you for bringing the mistake to our attention. The white phase in Figure 8d is a Zr-rich phase rather than a Y-rich phase. We have modified Figure 8d and modified the description related to Figure 8d. In addition, when the oxidation reaches 40 h, a Zr-rich phase appears in the inner oxide scale, which is the same as the oxide scale after oxidation for 100 h.

Round 3

Reviewer 2 Report

The manuscript can be recommended for publication in its current form.

English language is fine, minor editing is required.

Reviewer 4 Report

Authors answered my comments. The paper can be accepted for publication.